# Novel Approach to Task Scheduling and Load Balancing Using the Dominant Sequence Clustering and Mean Shift Clustering Algorithms

**Amer Al-Rahayfeh [1], Saleh Atiewi [1,\*], Abdullah Abuhussein [2] and Muder Almiani [3]**

[1]  Department of Computer Science, Al-Hussein Bin Talal University, Ma'an 71111, Jordan;
     amer.a.al-rahayfeh@ahu.edu.jo
[2]  Department of Information Systems, St. Cloud State University, St. Cloud, MN 56301, USA;
     aabuhussein@stcloudstate.edu
[3]  Department of Computer Information Systems, Al-Hussein Bin Talal University, Ma'an 71111, Jordan;
     malmiani@my.bridgeport.edu
\*   Correspondence: saleh@ahu.edu.jo; Tel.: +962-79-5160000

**Abstract:** Cloud computing (CC) is fast-growing and frequently adopted in information technology (IT) environments due to the benefits it offers. Task scheduling and load balancing are amongst the hot topics in the realm of CC. To overcome the shortcomings of the existing task scheduling and load balancing approaches, we propose a novel approach that uses dominant sequence clustering (DSC) for task scheduling and a weighted least connection (WLC) algorithm for load balancing. First, users' tasks are clustered using the DSC algorithm, which represents user tasks as graph of one or more clusters. After task clustering, each task is ranked using Modified Heterogeneous Earliest Finish Time (MHEFT) algorithm. where the highest priority task is scheduled first. Afterwards, virtual machines (VM) are clustered using a mean shift clustering (MSC) algorithm using kernel functions. Load balancing is subsequently performed using a WLC algorithm, which distributes the load based on server weight and capacity as well as client connectivity to server. A highly weighted or least connected server is selected for task allocation, which in turn increases the response time. Finally, we evaluate the proposed architecture using metrics such as response time, makespan, resource utilization, and service reliability.

**Keywords:** cloud computing; task scheduling; DSC algorithm; ranking; MHEFT algorithm; VMs; MSC algorithm; load balancing; WLC algorithm

## 1. Introduction

A growing number of organizations and industrial companies deploy their applications in cloud data centers due to the economy of scale provided by cloud computing [1]. To support the increased user demand, cloud computing service providers support ubiquitous services with pay-as-you-go payment methods. There are many challenges associated with the cloud environment. Task scheduling and load balancing are among the major challenges that are being faced by cloud adopters and are investigated by academia, research and industry [2].

Scheduling is a balancing scenario in which operations or tasks are scheduled according to the specific requirements and algorithm used. The purpose of scheduling algorithms in distributed load deployment systems is to optimize the processors and minimize the time of execution of the task. Job scheduling, one of the most popular improvement issues, plays a key role in improving flexible and reliable systems. The main purpose is to establish a schedule of posts on adaptable resources according

to adaptable time, which includes the identification of an appropriate sequence in which functions can be implemented within the framework of transaction logic constraints [3].

The remainder of this paper is divided into the following sections. Section 2, "Motivation", presents the main motivation for this research. Section 3, "Related Work", discusses previous task scheduling and load balancing methods. Section 4, "Problem Definition", highlights the problems in existing methods. Section 5, "Proposed Work", elucidates our proposed methodology. Section 6, "Performance Evaluation", presents the results and compares them with those of existing approaches. Section 7, "Conclusion", provides final explanations and future directions.

## 2. Motivation

This work is motivated by the following aspects:

- Most of the existing task schedulers doesn't consider the task's requirements, e.g., number of tasks and the task's priority, and some of them consider only the waiting time and response time reduction. [3] In this paper, we introduce a hybrid task scheduling algorithm that utilizes the shortest job first (SJF) and round-robin (RR) algorithms. The hybrid task scheduling algorithm comprises two major stages. First, the waiting time between short and long tasks is balanced. Waiting time and starvation are decreased via two sub-queues, Q1 and Q2. The combination in both versions of SJF and RR is evaluated with dynamic and static quanta, and optimality in the task scheduling and load balancing methods is achieved via evaluation. Second, the ready queue is divided into two sub-queues, in which Q1 denotes short tasks, whereas Q2 denotes long tasks [4].

- A good task scheduler considers environmental changes in its scheduling strategy [5]. This research proposes to use the ant colony optimization (ACO) algorithm for effectively allocating tasks to virtual machines (VMs). Slave ants adapt to diversification and reinforcement. The ACO algorithm avoids long paths with pheromones incorrectly accumulated by leading ants, thereby rectifying the ubiquitous optimization problem with slave ants [6].

- An NP-hard problem is a critical issue in cloud task scheduling. A metaheuristic algorithm can be used to solve this problem. A cloud task scheduling algorithm based on the ACO algorithm is proposed in this research to achieve efficient load balancing.

- The primary contribution of this research is minimizing the makespan time of a given set of tasks. ACO or a modified ACO algorithm is used to optimize makespan time [7]. The key aspects of cloud computing are task scheduling and resource allocation. A heuristic approach that combines the modified analytic hierarchy process (MAHP), bandwidth-aware divisible scheduling (BATS) + BAR optimization, longest expected process time (LEPT) preemption, and divide-and-conquer methods is adopted to schedule tasks and allocate resources. MAHP is used to rank incoming tasks. The BATS + BAR methodology is utilized to allocate resources to each ranked task. The loads of VMs are continuously checked with the LEPT method. If a VM has a large load, then other VMs are assigned tasks via the divide-and-conquer methodology [8]. The additional load is distributed across multiple servers using the load balancing strategy to optimize the performance of cloud computing. Load balancing issues are addressed via a hybrid bacterial swarm optimization algorithm. The particle swarm optimization (PSO) and bacteria foraging optimization (BFO) algorithms are combined—the former is for global search and the latter is for local search [9,10]. A critical factor in conflicting bottlenecks is to improve the response time for user requests on cloud computing. Accordingly, the throttled modified algorithm (TMA) is proposed in this research to improve the response time of VMs and enhance the performance of end to end users. Load balancing is performed via the TMA load balancer by updating and maintaining two index variables: Busy and available indices [10]. This research introduces a new and extensible VM migration scheduler to minimize completion time in scheduling. Live migration, which consumes network bandwidth and energy, is a high-cost scheduler. A migration scheduler computes the best moment for each migration and the amount of bandwidth to allocate

by relying on realistic migration and network models. The migrations that will be executed in parallel for fast migrations and short completion times are also decided by the scheduler [11].

This research proposes a hybrid load balancing algorithm that combines teaching-learning-based and gray wolf optimization algorithms to optimize load balancing for cloud computing. The scheduler allocates jobs to VMs in a distributed system and creates scheduling for resource allocation. In contrast with conventional optimization methods, the proposed method increases throughput [12]. Big data centers in clouds are provided with a dynamical load balance scheduling approach that increases network throughput and dynamically balances workload. This process implements two representative OpenFlow architectures, namely fully populated networks and fat-tree networks, which are dynamically migrated flows that require a high bandwidth in congested servers [13,14]. A stochastic load balancing scheme is used to reduce VM migration by considering the distance between source and destination physical machines [14]. VMs are migrated from heavily loaded physical machines to lightly loaded physical machines. The resource intensity-aware load algorithm that dynamically allocates various weights to different resources on the basis of their usage intensity is adopted, thereby decreasing the required time and cost and preventing unnecessary migrations via a strict migration algorithm [15].

This research primarily aims to improve response time through optimized task scheduling and load balancing algorithms, and its major contributions are as follows:

- The dominant sequence clustering (DSC) algorithm, which represents upcoming tasks as graphs, is adopted to schedule user tasks. More than one cluster exists in each graph. Metrics, such as deadline and makespan, are used to prioritize tasks and schedule them accordingly.
- The modified heterogeneous earliest finish time (MHEFT) algorithm, which schedules the highest priority task first for the subsequent process, is used to rank scheduled tasks.
- The mean shift clustering (MSC) algorithm, which clusters VMs in accordance with a kernel function, is utilized to cluster VMs.
- The weighted least clustering (WLC) algorithm, which provides weight to each server on the basis of their capacity and client connectivity, is adopted to balance loads. Tasks are allocated by a highly weighted server, which increases response time.

## 3. Related Work

Load balancing has been extensively studied in the literature, and many algorithms have been proposed. Authors in [16] proposed a two-stage strategy for enhancing the performance of task scheduling and decreasing unnecessary task allocation in clouds. The Bayes classifier principle is first adopted for classifying tasks based on historical scheduling data, and then tasks are dynamically coordinated with corresponding concrete VMs. However, the Bayes classifier requires considerable historical data to obtain accurate results, and consequently, this strategy leads to accuracy loss. The research reported in [17] introduced a priority-based task scheduler that ranks users on the basis of task length and memory. A hybrid optimization algorithm that integrates genetic and PSO algorithms for efficient task scheduling was also proposed. A queue manager is utilized to store and prioritize user tasks. Rejected tasks from the prioritized queue are given to the on-demand queue, and tasks from both queues are given to the hybrid algorithm. Nevertheless, slow convergence rate and poor local search capability are observed from the PSO algorithm. In [18], a hybrid optimization algorithm was suggested that combines cuckoo and harmony search algorithms to develop an intelligent method for enhancing the scheduling process. Tasks are initially assigned to VMs, and fitness functions, which consider metric energy consumption, memory usage, credit, and penalty, are calculated. The cuckoo and harmony search algorithms are used for updating, and the best fitness function is selected through hybridization. However, slow local search convergence, due to the harmony search algorithm, limits the method. Authors in [19] proposed two hybrid algorithms, namely a combination of fuzzy logic and PSO algorithm and a combination of simulated annealing and PSO algorithm, for improving

task scheduling performance in cloud computing. A waiting time optimization algorithm based on the PSO algorithm is used to minimize task waiting time. The objectives of the two hybrid algorithms are to utilize resources, optimize performance metrics, minimize makespan, and achieve good load balancing. Nevertheless, the two hybrid algorithms require considerable computational time. The researchers in [20,21] introduced dynamic voltage and frequency scaling that describes the quality-of-service requirements of tasks with minimum frequency. The energy consumption ratio is used and evaluated to determine different frequencies. Upcoming tasks are distributed to active servers. After a task is completed, the remaining tasks are rescheduled among processors via a processor-level migration algorithm. Runtime optimization is also analyzed, and a task migration and frequency readjustment scheme are provided to decrease the energy consumption of the server. Reference [21] further proposed a dynamic heterogeneous SJF model for minimizing actual Central Processing Unit (CPU) time and overall system execution time or makespan. Energy consumption is decreased via dynamic provisioning, and the dynamic request is matched with dynamic heterogeneity, thereby reducing makespan and increasing resource utilization. The dynamic heterogeneities of workload and resources are considered. Dynamic requests for heterogeneous resources are defined to minimize execution time and increase resource utilization. Reference [22] aimed to solve the problem of dynamic load balancing by proposing a hybrid PSO algorithm. Workload is distributed across different resources via load balancing. The multiple kernel function support vector machine (SVM) algorithm is used to evaluate the disturbances in utilizing various resources. A centralized cloud-based multimedia system that comprises a resource manager, cluster heads, and server clusters is considered. Client requests are assigned by the resource manager to the servers within its cluster head. However, the training time of the SVM algorithm is slow. Reference [23] presented a load balancing method based on constraint measure. The capacity and load of each VM are initially computed. If the load is larger than the threshold value, then the load balancing algorithm allocates tasks to the VMs. The deciding factor of VMs is calculated, and the loads of VMs are checked by the algorithm. The selection factor for each task is then calculated, and the task with a better selection factor than those of others is allocated to the VMs. Reference [24] proposed the raven roosting optimization algorithm that uses an efficient load balancing method to solve the task scheduling problem, where roosts represent information centers. VM capacity is determined to transfer overloaded tasks. References [25,26] introduced the deadline-aware priority scheduling model, which minimizes the average makespan and maximizes resource utilization under deadline constraint. Tasks are scheduled in ascending order based on length priority and matched to the suitable VM with minimum processing time. Reference [27] proposed the uncertainty-aware online scheduling algorithm for enhancing the performance of cloud service platforms. The count of tasks that directly wait on VMs is controlled, and proactive and reactive strategies are combined. The feature parameters of applications are computed after they are submitted, and tasks are allocated to service instances for execution. Reference [28] improved the energy efficiency of heterogeneous servers in a cloud computing system via a noncooperative game-based model. The model represents server utility function as unit power efficiency and proves the existence of the Nash equilibrium point of game. The total request amount in the system serves as a constraint, and the appropriate task quantity is selected by each server that matches it with the optimum resource to handle user request. The Nash equilibrium is solved by a proposed Lagrange multiplier-based distributive iterative algorithm, but the shadow values of this algorithm lead to the low accuracy of the model. Reference [29] provided a deadline-aware PSO algorithm for optimizing the performance of task scheduling algorithms. Two parameters, namely deadline and profit, are utilized by the algorithm and calculated to classify tasks. Each task is scheduled to VMs on the basis of priority using the PSO algorithm. If two tasks have the same deadline values, then the maximum user pay value, which considers deadline and million instructions, is computed. Tasks are scheduled on the basis of value. However, the PSO algorithm exhibits low convergence time.

## 4. Problem Definition

A set of users (u1, u2, . . . , un) and their tasks (t1, t2, . . . , tn) are considered in a cloud system. The primary objective of this research is to improve the following metrics for each user task:

(a)   Response time;
(b)   Makespan time;
(c)   Resource utilization;
(d)   Service reliability.

In [1], the temporal task scheduling algorithm (TTSA), which dispatches an entire task into a private CDC and public clouds, was used to schedule tasks. A hybrid simulated annealing-PSO is used to solve the cost minimization problem in each iteration of TTSA. The throughput can be increased, the cost of a private CDC can be decreased, and the delay bounds of all tasks can be met by TTSA. However, the convergence speed of simulated annealing is slow, and computation time is considerable. In [2], the multi-rumen anti-grazing approach was used to balance loads. This approach comprises three phases. In phase one, VM capacity and task size are used to calculate the task completion time (TCT) with a communication delay matrix. In phase two, tasks are assigned to respective VMs, and makespan and TCT values are updated for every iteration. In phase 3, tasks are rescheduled into new VMs in accordance with the new TCT and makespan values. Task rescheduling increases computational time and complexity. In [4], a hybrid of SJF and RR schedulers that considers only dynamic variable quantum tasks is adopted to complete task scheduling. The ready queue is divided into Q1 and Q2, where Q1 is for storing short tasks, whereas Q2 queue is for storing long tasks. The burst time for each task is acquired, which determines short and long tasks, and used to identify the median value that separates tasks. Waiting and response times are shortened in this process, but computation time for large tasks is increased due to the use of dynamic quantum computation to calculate the median value. In [9], a hybrid algorithm that combines BFO and PSO algorithms was used to balance loads. Jobs are collected as batches, and specific resources are allocated for each job. Makespan, operational cost, and utilization are calculated after the resource allocation process. The obtained values are used to schedule jobs to appropriate VMs. Slow convergence time and poor local search capability caused by the PSO algorithm are determined. In the current research, we use efficient task scheduling and load balancing algorithms to solve the problems highlighted in [1,2,4,9].

## 5. Proposed Work

### 5.1. System Overview

Task scheduling and load balancing algorithms are used in the proposed work to improve response time. The architecture of the proposed work, in which we use the DSC algorithm to schedule tasks on the basis of their priorities, is presented in Figure 1. Tasks are prioritized in consideration of deadline and makespan. Tasks are rescheduled and ranked to determine the highest priority task, which is scheduled first for upcoming processes. VMs are clustered on the basis of the kernel function using the MSC algorithm. The WLC algorithm, which considers the capacity and client connectivity of servers, is used to balance loads, and then tasks are allocated to VMs.

The flowchart of our proposed work indicates that the incoming tasks of users are directly given to the DSC algorithm, and the MHEFT algorithm is used to rank scheduled tasks. The highest priority task is transferred to the VM allocation process, and the MSC algorithm is used to cluster VMs. The WLC algorithm, which considers server capacity and client connectivity, is used for load balancing. Weights are subsequently allocated to each VM, and tasks are allocated to VMs with the highest weight value.

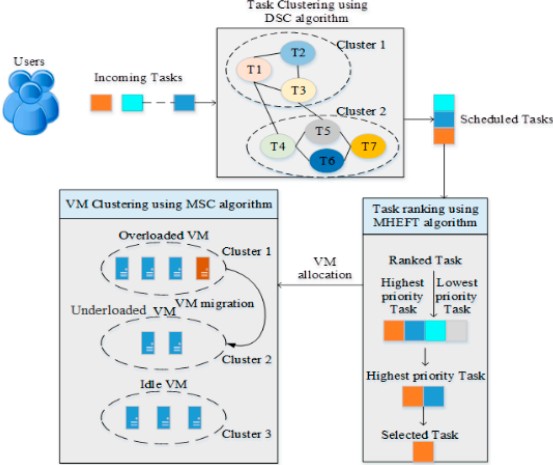

**Figure 1.** Architecture of the proposed work.

*5.2. Task Scheduling*

The DSC algorithm is used to schedule tasks. The priority of each upcoming task is considered for scheduling and clustering.

5.2.1. Task Clustering

The DSC algorithm clusters upcoming tasks by computing priority (P (i)) values using the top (t (i)) and bottom (b (i)) levels of the tasks. The top level (t (i)) of a task is the length of the longest path from an entry task to i. The bottom level (b (i)) is defined as the length of the longest path from i to an exit task. Task priority is computed as

$$P(i) = t(i) + b(i) \tag{1}$$

where $P(i)$ is the priority of task $i$, $t(i)$ is the top level of task $i$, and $b(i)$ is the low level of task $i$. The top and low levels of a task are summed up to compute its priority *P(i)*.

Task clustering and scheduling using the DSC algorithm are illustrated in Figure 2. Priority values are computed for each task and used in task clustering, followed by task scheduling.

Scheduling based on the priority of each task is depicted in Figure 3, where cluster 1 denotes priority 1 tasks, and cluster 2 denotes priority 2 tasks. Equation 1 is used to prioritize tasks. Tasks are clustered on the basis of their priorities via the DSC algorithm. In the preceding example, tasks 4 to 7 serve as priority 1 and tasks 1 to 3 serve as priority 2 based on their deadline and makespan. Scheduled tasks are ranked to select the highest priority task to be transferred to the upcoming processes.

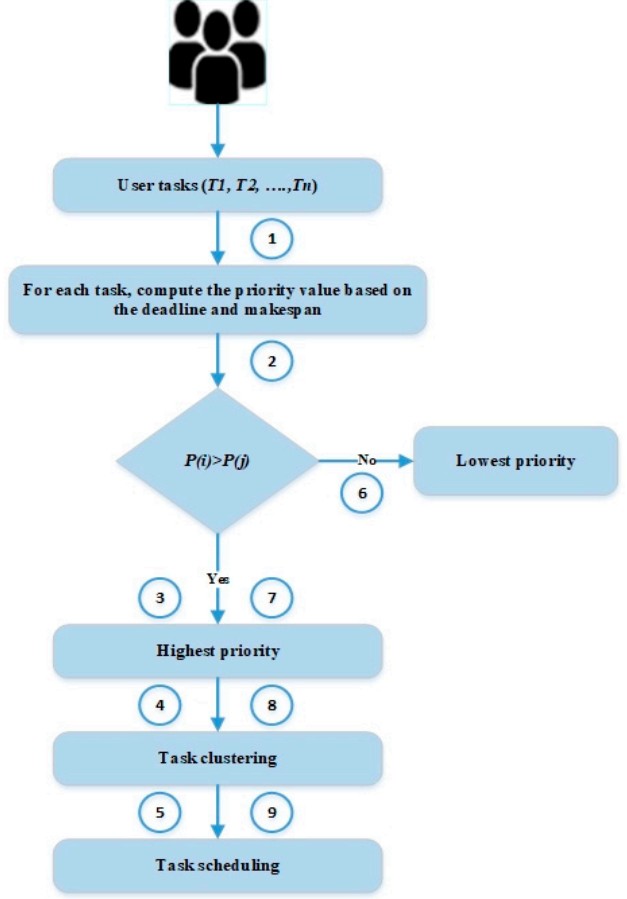

**Figure 2.** Task clustering and scheduling.

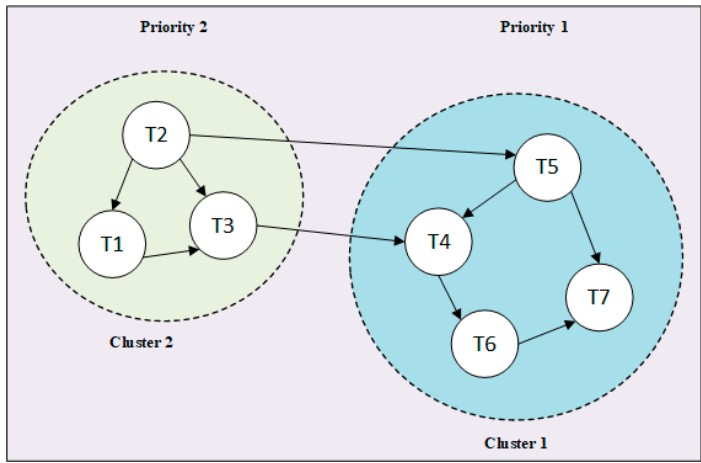

**Figure 3.** Task clustering.

### 5.2.2. Task Ranking

The MHEFT algorithm is used to rank scheduled tasks generated using the DSC algorithm. The highest priority task among the scheduled tasks is selected for the upcoming processes, such as VM clustering and load balancing. The average execution and communication times for transferring data are calculated to rank tasks.

MHEFT, which is a variant of the static list algorithm that prioritizes tasks with considerable influences on total workflow execution time, is the most efficient ranking algorithm. Procedures

that are static methodologies for mapping issues and that accept static condition for a given time are utilized. The upward rank of each task, which is defined as the critical path of the task and considers the maximum communication time and average implementation time from the start to the end of any task, is calculated to rank the tasks. If two tasks have the same priorities, then arrival time is checked, and first priority is given to the task that arrives first. The rank of task $R(t_i)$ is expressed as

$$R(t_i) = E(t_i) + \max_{t_j = s(t_i)} \left( c_{i,j} + R\left(t_j\right) \right) \tag{2}$$

where $E(ti)$ is the average execution time of the task across all VMs, $S(ti)$ is the set of immediate successors of the task, $C_{i,j}$ is the communication time that corresponds to the transfer of data $i,j$ via edges $i$ and $j$, and $R(tj)$ is the computation of all of its children. Communication time $C_{i,j}$ is computed using

$$C_{i,j} = y + \frac{data_{i,j}}{b} \tag{3}$$

where Y is the average latency, and b is the average bandwidth of communication links among VMs in the system. We can rank each scheduled task by computing Equations 2 and 3. The highest priority task is selected and transferred to the VM allocation process for further operations. Expression ranking and the selection of the highest priority task are completed via the MHEFT algorithm.

### 5.3. Load Balancing

The WLC algorithm is used for load balancing. The MSC algorithm initially clusters VMs, thereby reducing VM migration. Load is then balanced, and tasks are allocated to the optimum VM for increasing response time.

### 5.3.1. VM Clustering

The MSC algorithm is used to cluster VMs, and unnecessary VM migration in the process is consequently reduced. This nonparametric clustering technique does not require prior knowledge and shape of clusters, locates the maximum density function, and is consequently known as a mode-seeking algorithm. It is based on the sliding window algorithm, which aims to find the dense area of data points. The MSC algorithm is a centroid-based algorithm with the primary objective of locating the center points of each group. It works by updating candidate center points within the sliding window. Nearby duplicates of candidate windows are eliminated by filtering, and the final set of center points and their parallel groups are then formed. Input is considered data points in the MSC algorithm.

Given "n" data points di, where i = 0, ... , n on a d-dimensional space, a multivariate kernel density function is estimated with kernel $K(x)$ as

$$F(x) = \frac{1}{nr^d} \sum_{i=1}^{n} K\left(\frac{x - x_i}{r}\right) \tag{4}$$

where $r$ denotes the sliding window radius. For radially symmetric kernels, the profile of kernel $k(x)$ is defined and satisfies the following equation,

$$K(x) = e_{k,d} k(\|x\|^2) \tag{5}$$

where $e_{k,d}$ refers to the normalization constant that assures that $K(x)$ is integrated into 0. The modes of density functions are located at the zero of the gradient function $\nabla F(x) = 0$,

$$\begin{aligned}
\nabla F(x) &= \frac{2e_{k,d}}{nr^{d+2}} \quad \sum_{i=1}^{n} (x_i - x) \, g\left(\|\tfrac{x - x_i}{r}\|^2\right) \\
&= \frac{2e_{k,d}}{nr^{d+2}} \left[ g\left(\|\tfrac{x - x_i}{r}\|^2\right) \right] \left[ \frac{\sum_{i=1}^{n} x_i \, g\left(\|\tfrac{x - x_i}{r}\|^2\right)}{\sum_{i=1}^{n} g\left(\|\tfrac{x - x_i}{r}\|^2\right)} - x \right]
\end{aligned} \tag{6}$$

where $g(s) = -k'$ (s). The first term is proportional to the density estimate at $x$ calculated with kernel $G(x) = e_{k,d} \, g(\|x\|^2)$, and the second term is the mean shift. The mean shift vector constantly approaches the direction of maximum increase in density. The following vector computations are performed to realize the mean shift procedure:

- Computation of mean shift vector $M_r(x^t)$;
- Translation of window $x^{t+1} = x^t + M_r(x^t)$.

The mean shift vector is expressed as:

$$M_r\left(x^t\right) \frac{\sum_{i=1}^{n} x_i \, g\left(\|\frac{x-x_i}{r}\|^2\right)}{\sum_{i=1}^{n} g(\|\frac{x-x_i}{r}\|^2 \, )} - x \tag{7}$$

Mean shift is a hill-climbing algorithm in which the kernel for each iteration shifts to a higher-density region until it converges. The mean shift defines every shift in this algorithm. The kernel approaches the centroid or mean of the points within it in every iteration. The density within the sliding window is equal to the number of points within it. Kernel function selection determines the mean computation. The sliding window is shifted until convergence, where no direction at which shift can occupy points inside the kernel.

Clustering via the MSC algorithm is illustrated in Figure 4, which shows that the kernel function approaches the highest density area in each iteration. VMs are clustered, and tasks are optimized to increase response time.

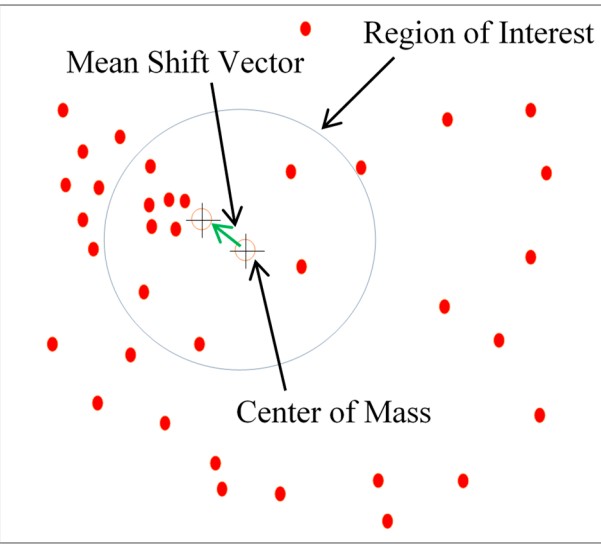

**Figure 4.** Example of the MSC Algorithm.

5.3.2. VM Allocation

The WLC algorithm is used to balance phase loads. This algorithm weighs each server on the basis of their capacity and client connectivity. Tasks are allocated by the server with the highest weight. The weight of each VM is computed using the WLC algorithm. Parameters, such as CPU speed, idle rate, memory size, and input/output (I/O) capacity, define server capability. We compute the weight factor for each VM by utilizing these parameters and express it as follows,

$$G_i = \sqrt{(\rho_m \times H_m \times D_m) + (\rho_c \times S_c \times D_c)} \tag{8}$$

where $D_m$ is the idle rate of the node memory; $D_c$ is the idle rate of the node CPU; $H_m$ is the memory size; $\rho_m$ and $\rho_c$ are proportional constants, and their summation is always equal to 1, i.e., $\rho_m + \rho_c = 1$;

and $S_c$ is the CPU speed (MHz). The weights of each task are determined using Equation 8, and the weight load of individual VM is calculated using the ratio of the number of tasks in the VM to the execution time of the tasks, i.e.,

$$L = \frac{N(t)}{E(t)} \tag{9}$$

where *N(t)* is the number of task, and *E(t)* is the execution time of each task within the VM. Tasks are allocated in consideration of the highest weight and minimum load of each VM. VMs are used to allocate the highest priority task generated by the MHEFT algorithm. We increase the response time of each task using the aforementioned approach for load balancing and task allocation.

## 6. Performance Evaluation

Our experimental results are presented in this section. The proposed work that aims to improve response time for user task is compared with an existing method. The simulation setup is discussed in Section 6.1, performance metrics discussed in Section 6.2 and a comparative analysis is provided in Section 6.3.

### *6.1. Simulation Setup*

Our proposed work is implemented in CloudSim 3.0 simulator environment. CloudSim has become one of the most popular open source cloud simulators in research and academia. CloudSim is completely written in Java. CloudSim models and simulates extensible clouds and supports cloud system components, such as VMs and data centers. Various functionalities and exceptional characteristics, such as the generation of different workloads with various scenarios, are provided, and a robust test is performed via customized configurations. The hardware and software requirements of our proposed work are presented in Tables 1 and 2, respectively.

**Table 1.** Hardware requirements.

| Component | Specification |
|---|---|
| Operating system | Windows (X86 ultimate) 32-bit OS |
| Processor | Intel®Pentium®CPU G2030 @ 3.00 GHZ |
| RAM | 2.00 GB |
| System type | 32-bit OS |
| Hard disk | 500 GB |

**Table 2.** Parameters required in CloudSim.

| Entities | Specifications | Ranges |
|---|---|---|
| Cloudlets/tasks | Total number of tasks | 15–200 |
| | Task length | 1500–3000 |
| VM | Host | 3 |
| VM/physical machine | Storage | 300 GB |
| | Bandwidth | 200,000 |
| | Memory | 520 |
| | Butter capacity | 20 |
| | MIPS/PE | 400 |
| | Bandwidth cost | 0.2/MB |

Table 1 lists the hardware requirements of our proposed work. Windows (X86 ultimate) 32-bit OS is used to operate our proposed work, and a 2.00 GB RAM is utilized for the hardware system.

Table 2 provides the parameters used for the implementation process of our proposed work. Tasks within the range of 15–200 are selected.

*6.2. Performance Metrics*

Four performance metrics are considered to evaluate and compare the performance of our proposed work with that of an existing system.

6.2.1. Response Time

The response time of a task refers to the time intervals among tasks to arrive into the system until its completion. Response time $R_e$ is expressed as

$$R_e = T_c - T_a + T_t \tag{10}$$

where $T_c$ represents the time required to complete a task, $T_a$ represents the arrival time of a task, and $T_t$ represents the transfer time of a task.

6.2.2. Makespan

Makespan is defined as the total time taken to process a set of tasks for its complete execution. Makespan $M_a$ is represented as

$$M_a = max(CT) \tag{11}$$

where *max(CT)* is the maximum time required to complete all tasks.

6.2.3. Resource Utilization

Resource utilization denotes the number of resources required during task execution. Resource utilization $R_u$ is expressed as

$$R_u = \frac{T_c}{M_a \times N} \tag{12}$$

where $T_c$ represents the time taken to complete a task, $M_a$ represents makespan and $N$ represents number of resources.

6.2.4. Service Reliability

Three parameters, namely accessibility, continuity, and performance, are considered in service reliability in cloud computing. Accessibility refers to service availability whenever a customer requires service. Continuity refers to the non-disturbance of services over a particular duration. Performance refers to the complete realization of customer expectation. The service reliability metrics in our proposed work consider the three parameters and are compared with those of an existing method.

*6.3. Comparative Analysis*

We compare the performance metrics of our proposed work, namely response time, makespan, resource utilization, and service reliability, with those of an existing method.

6.3.1. Response Time

The primary objective of our proposed framework is to reduce response time. A comparison was conducted between the response times of our proposed work and TTSA. Response time refers to the time that elapsed among task arrivals until complete execution of the tasks. The simulation result indicates that the response time of our proposed work is less than that of the TTSA method.

The comparison of the response time between the proposed and existing methods is depicted in Figure 5. For a user with size 20 kB, our proposed work achieves a task response time of 15 s, whereas the TTSA method has response time of 20 s. For a 40 kb user, the response time of our proposed work is 10 s, whereas that of TTSA is 15 s. Therefore, the proposed work has a shorter response time than that of the existing method, which is realized via VM clustering. VM migration is reduced, and tasks are allocated and processed rapidly, thereby decreasing response time.

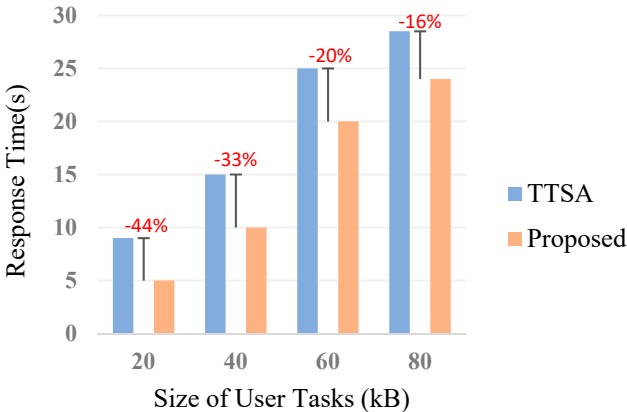

**Figure 5.** Comparison of response time.

### 6.3.2. Makespan

A comparison of makespan, which is the total time taken to execute a set of tasks, is also performed. The simulation result indicates that our proposed work has a shorter makespan time than that of TTSA.

The comparison of makespan between the proposed and existing methods is presented in Figure 6, in which the finishing time of the final task does not contain any overhead in our proposed work. VMs are provided with a light load and high computing power for executing several tasks in different sizes via the load balancing algorithm of WLC, which balances load on the basis of the weight assigned to each VM. The highest priority task is allocated first to the highest weight VM. This method increases the efficiency of the VM in executing various tasks, and thus, decreases the makespan time of our proposed work compared with that of TTSA.

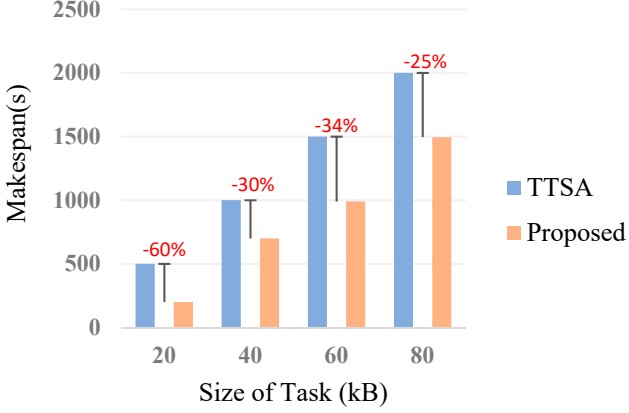

**Figure 6.** Comparison of makespan.

### 6.3.3. Resource Utilization

Resource utilization is increased through task clustering using the DSC algorithm. Deadline and makespan are considered to classify each upcoming task via prioritization. Figure 7 presents the comparison of resource utilization.

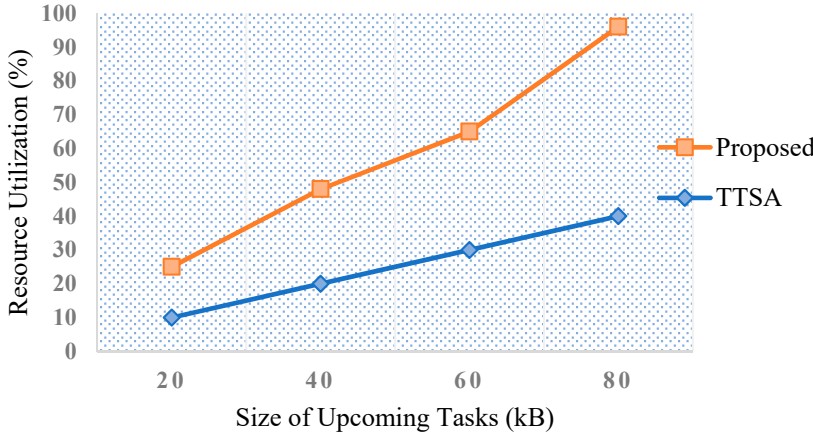

**Figure 7.** Comparison of resource utilization.

The resource utilization of each task depends on its requirement. When upcoming tasks utilize the required resources, the resource utilization of our proposed method increases. Resources comprise CPU, I/O, and memory. Each VM represents resource parameters as $D_c$, $S_c$, and $D_m$, which refer to CPU idle rate, CPU speed, and memory idle rate, respectively. The DSC algorithm is used in our proposed work to cluster upcoming tasks on the basis of their priorities obtained by computing their deadline and makespan. Each prioritized task is then ranked using MHEFT, and the highest priority task is selected. Our proposed method increases resource utilization metrics via clustering and highest priority task selection by up to 98% compared with that of the existing method.

### 6.3.4. Service Reliability

The service reliabilities of the proposed and existing methods, which include performance, accessibility, and continuity, are also compared. Efficient task scheduling and load balancing improve the service reliability of our proposed method. In our framework, resource utilization is enhanced by task clustering and ranking, whereas load balancing is improved via VM clustering and allocation, which reduce VM migration. The response time of our proposed work is accordingly reduced, and service reliability is increased. The service reliability of a task is presented as a plot between task size and trust level. Cloud computes the trust value of each user-given task in cloud environment on the basis of which service is provided to the upcoming user task.

Figure 8 shows that our proposed work exhibits greater service reliability than TTSA.

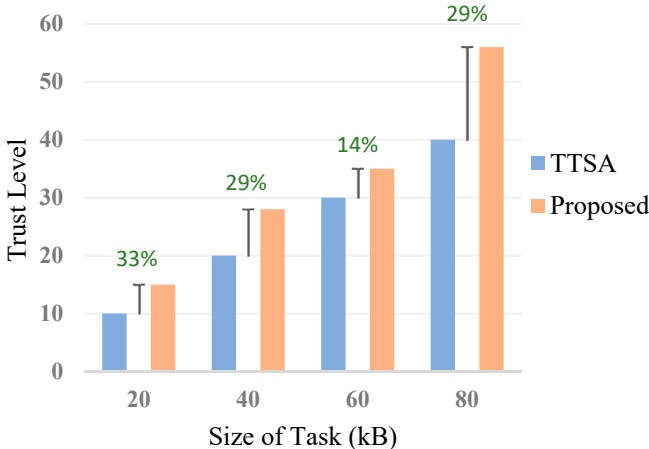

**Figure 8.** Comparison of service reliability.

## 7. Conclusions

This research proposes a novel approach for task scheduling and load balancing using the DSC and MSC algorithms to increase response time for user task. Incoming tasks are clustered using the DSC algorithm on the basis of priority. Priority value is computed according to the top and bottom levels of the tasks and is used to form clusters for task scheduling. The MHEFT algorithm is adopted to rank scheduled tasks via rank value computation, in which the average execution and communication times for each task are considered. The highest priority task is selected and assigned first for upcoming processes. Resource utilization in our work is increased by such task clustering and ranking compared with that in existing methods. In load balancing, the MSC algorithm, which is a hill-climbing algorithm that shifts the kernel for each iteration to a higher-density region until convergence, clusters VMs. The mean shift defines every shift in this algorithm, and the kernel approaches the centroid or mean of the points within it in every iteration. The kernel that approaches the highest density function serves as the basis for computing the mean shift vector. The WLC algorithm, which assigns weight to each VM on the basis of its client connectivity, CPU idle rate, and CPU speed, is used for load balancing after clustering. The weight value for each VM in a cluster is computed by considering CPU idle rate, CPU speed, memory idle rate, and memory size. The highest weight VM is selected for task allocation, which reduces the response time of the user task. Response time, makespan, resource utilization, and service reliability are considered in our proposed work. The simulation results indicate that the performance metrics of our proposed work are better than those of the TTA method, particularly in decreasing response time. Our proposed method realizes 98% resource utilization, which is higher than that of TTSA. In the future, we plan to integrate a new intelligent optimization algorithm to select the highest priority task and to adopt an efficient load balancing algorithm to improve execution time in our future work.

**Author Contributions:** Conceptualization, A.S. and R.A.; methodology, A.S.; software, A.S.; validation, A.A., A.M.; formal analysis, A.A.; investigation, R.A.; resources, A.S.; writing—original draft preparation, A.S.; writing—review and editing, A.A.; visualization, A.M.; supervision, A.S.; project administration, R.A.

**Funding:** This research received no external funding.

**Acknowledgments:** The authors thank the anonymous reviewers for their insightful comments that helped improve the quality of this study.

**Conflicts of Interest:** The authors declare no conflict of interest.

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
