# Peer review of "Novel Approach to Task Scheduling and Load Balancing Using the Dominant Sequence Clustering and Mean Shift Clustering Algorithms"

_futureinternet, doi:10.3390/fi11050109_

Round 1
Reviewer 1 Report
Please revise the paper based on the following comments and suggestions:
The figures have really bad quality. The figures needs to be improved, and additionally, please revise the use of the figures.
The proposed work is not well defined. It is not clear how the algorithm work and which is its significance.
The results are not well explained and the observations and contribution of the proposed work are not clear.
The equations are not well explained.
The conclusion is weak and the reviewer don't see any "superiority" statement regarding the proposed work.
Author Response
Please see the enclosed file
Regards

Reviewer 2 Report
A review of English should be conducted in the scientific paper to avoid ambiguities. Reconfirm the following editorial concerns according the author's guidelines for readability. Please editorial check the whole text and figures (letter size, description etc.) again.
Author Response
Please see the enclosed file
Regards

Reviewer 3 Report
How I read this paper
Novel Approach to Task Scheduling and Load Balancing Using the Dominant Sequence Clustering and Mean Shift Clustering Algorithms
Cloud Computing (CC) à
Shortcomings / motivation à
- Most task schedulers do not consider task requirements (e.g. number of tasks and prio) and some only consider waiting time and response time
o SJF and RR algorithms included
- Account for environmental changes
o Ant Colony algorithm (ACO) to allocate tasks effectively to VM’s
o NP-hard problem (define?)
- Minimizing makespan time of set of tasks
Steps to prove own superior algorithm:
1) Task Scheduling à Dominant Sequence Clustering (DSC) MHEFT, and MSC algorithm
2) Load balancing à Weighted Least Connection (WLC) algorithm
3) Evaluate on criteria response time, makespan, resource utilization, service reliability
In detail;
1) Tasks clustered using DLC algorithm (user tasks as graph of one or more cluster)
2) Rank using Modified Heterogeneous Earliest Finish Time (MHEFT) algorithm (highest prio first)
3) Virtual Machine (VM) clustering using Mean Shift Clustering (MSC) using kernel functions
4) Load balancing using WLC algorithm (distribute load based on server weight and capacity as well as client connectivity to server à highly weighted or least connected server selected for task allocation which increases response time
5) Evaluate proposed architecture using metrics such as response time, Makespan, Resource Utilization, Service reliability
Supported with flowchart, mathematical calculations and supporting results.
Review
Overall: consider where and where not to use ‘the’.
L19: cluster à plural?
L32-33: consider rephrasing [1]
L34: on rental basis
L40 – L41: clarify on terminology >> Job = task = posts = operational scheduling (all same?)
L51 – L108 – Motivation
Motivation is a somewhat misplaced term here. This is a combination of what can be done better in task scheduling and load balancing, but also introduced underlying algorithms used for the calculations further in the paper. For clarift purposes, it would be better to make a more solid motivation, specifically aimed at what the DLC, MHEFT, MSC algorithms would solve. Perhaps consider moving the mentioned algorithms / calculation setups to the below (to support the calculations) for more structure. E.g. where would do the mentioned SJF and RR algorithms support and where are they incorporated / used in the novel approach?
L109 – L120: good summary! Linkage to motivation above can be improved, as mentioned. This is the core.
L70 typo ‘ressearch’
L121: Related work
Insightful, but what about relevance? Are there any lessons incorporated from the related work? Why relevant for use in DLC / MHEFT / MSC algorithms?
L324: consider enter / next page
L370-371: Subsection 5.1 and 5.2 -> 6.1, performance metrics 6.2 and comparative analysis in 6.3
Further: Clear setup and results structure.
Author Response
Please see the enclosed file
Regards

Reviewer 4 Report
The proposal of the paper is interesting and fits the scope of the Journal. Nonetheless, the manuscript requires extra efforts to improve its quality and presentation for the prestigious journal Future Internet. After a careful revision, a set of comments are expounded hereafter.
- The manuscript is well written and organized. However, there are some minor mistakes or improvements to make regarding the format of the document, as commented below.
“and” must be inserted between the names of the last two authors.
The acronym IT, Information Technology, should be decomposed the first time it is used. The same comment is applied to CPU (line 152).
In line 34, “cloud providers provide” must be revised in order to change the verb by another one with the same meaning, avoiding redundancies.
In the fourth section, the metrics are enumerated; however, they should start with capitalized letter in order to be coherent with the subsection 6.2.
The titles of tables lack the terminal period (punctuation).
In line 511, the word future should not be capitalized.
Concerning the references, they must be thoroughly revised according to the template. For instance, the abbreviated names of journals must be used.
- About the content of the manuscript, the comments after a careful revision are the following:
In the third section, the structure “Reference [x]” is used repeatidly, which should be improved. Other statements can be used alternatively, for instance “The research reported in [x]” or “Author1 et al. [x] suggested…”. This way, the readability of the manuscript will be enhanced.
The literature review has been conducted in an exhaustive manner.
In Section 5, a sentence before the first subsection would be convenient in order to briefly comment subsections that compose the whole section. This will help the reader to understand the structure of the paper.
In Section 6, the open-source nature of the CloudSim environment should be mentioned since it is a desirable feature for cloud-related resources, from this humble reviewer viewpoint.
In Figure 5, the units indicated for the horizontal axis must be revised, namely, “kb” should be replaced by “kB”. The same occurs for the text in line 442 and for the figures 6 and 8.
In the Conclusion section, mentioning some limitation of the conducted research would be interesting for the reader and would enhance the presentation of the manuscript.
As a conclusion of the revision, in its current state, the manuscript must address the provided suggestions to reach a better presentation and scientific level, according to the prestigious journal Future Internet.
Author Response
Please see the enclosed file
Regards
